# Use, Practices and Attitudes of Sports Nutrition and Strength and Conditioning Practitioners towards Tart Cherry Supplementation

**DOI:** 10.3390/sports9010002

**Published:** 2020-12-22

**Authors:** Vlad Sabou, Jimmy Wangdi, Mary F. O’Leary, Vincent G. Kelly, Joanna L. Bowtell

**Affiliations:** 1Sport and Health Sciences, St Luke’s Campus, University of Exeter, Heavitree Road, Exeter EX1 2LU, UK; vs348@exeter.ac.uk (V.S.); j.wangdi@uq.net.au (J.W.); M.OLeary@exeter.ac.uk (M.F.O.); 2School of Human Movement and Nutrition, University of Queensland, Brisbane, QLD 4072, Australia; 3School of Exercise and Nutrition Sciences, Queensland University of Technology, Brisbane, QLD 4000, Australia; v6.kelly@qut.edu.au

**Keywords:** athletic performance, polyphenols, ergogenic aids, exercise recovery, sleep, sports

## Abstract

Tart cherry (TC) supplementation has been shown to accelerate post-exercise recovery, enhance endurance performance and improve sleep duration and quality. This study aimed to identify the use, practices and attitudes of sports nutrition and strength and conditioning practitioners towards tart cherry supplementation. Thirty-five practitioners anonymously completed an online survey investigating their use, practices and attitudes towards tart cherry supplements. Forty-six percent of the responders were currently recommending TC supplements, 11% had previously recommended TC supplements and 26% have not previously recommended TC supplements but were planning on doing so in the future. Of those recommending TC, 50% recommended or were planning on recommending TC supplements to enhance exercise recovery and 26% to improve sleep duration and quality. Acute supplementation and daily use during multi-day competition or demanding training blocks with a 2–3-day pre-load were the most reported supplementation recommendations (28% and 18%, respectively). Fifty-two percent of responders indicated uncertainty about the daily polyphenol dose to recommend as part of a TC supplementation protocol. Despite the high use and interest from sports nutrition and strength and conditioning practitioners in TC supplements, their practices did not align with the protocols found to be effective within the literature.

## 1. Introduction

Tart cherries (TC) are a rich source of phytochemicals, containing amongst other compounds a complex mix of polyphenols, including phenolic acids and several flavonoid subgroups [1]. Tart cherry supplements have been shown to exert antioxidant and anti-inflammatory effects [2,3,4]. These properties are of relevance to athletes and coaches given the potentially negative effects of oxidative stress and inflammation on various aspects of athletic performance [2,5]. Multiple studies support the beneficial effects of TC supplementation on recovery from several exercise modalities including strenuous resistance training [4,6], endurance running [2,7] and intermittent running/sprinting protocols [8,9,10]. Positive effects on sleep duration and quality have also been identified following TC supplementation in a limited number of studies [11,12]. Lastly, emerging evidence illustrates the potential for both acute and chronic TC supplementation to enhance endurance performance [13,14]. These findings support several applications for TC supplementation in various athletic environments, including intermittent, endurance and power and strength-based sports.

The use of dietary supplements amongst athletes is widespread with previous studies indicating that 51–87% of professional athletes include them as part of their nutritional strategy [15,16,17]. Nevertheless, the efficacy of dietary supplements varies substantially and depends on the implementation of an effective supplementation protocol. Previous research shows that applied practice does not always reflect research-based protocols and recommendations for sports nutrition supplementation strategies [18,19]. Division I collegiate athletes utilising creatine supplements, and Australian professional footballers utilising β-alanine supplements did not follow evidence-based supplementation protocols [18,19]. However, when athletes receive nutritional counselling, their dietary supplementation practices appear to be improved [20]. Several recent studies identify sports nutritionists, sports dietitians, athletic coaches, strength and conditioning coaches and exercise physiologists as the main providers of nutrition support and guidance for elite athletes [19,21,22]. It is important that these practitioners are aware of the current recommendations for dietary supplement use to ensure that their athletes are informed on evidence-based supplementation protocols.

Whilst there are several applications for TC supplements in athletic environments, little is currently known about the prevalence of use, or the practices and attitudes of sports nutrition and strength and conditioning practitioners towards TC supplementation. Akin to β-alanine and creatine [18,19], the use of TC supplements in applied practice may differ from research-based protocols. Indeed, there are differences in supplementation protocols between published ‘applied’ [23,24] and ‘laboratory-based’ TC research studies [4,6,8] and less evidence of ergogenic effects in applied studies. This suggests that protocols in applied practice may deviate from laboratory-based protocols, at the risk that the ergogenic value of TC supplementation may be compromised.

This study therefore aims to investigate the prevalence of use, practices and attitudes of sports nutrition and strength and conditioning practitioners towards TC supplements, and to compare them with the current body of literature in order to inform best-practice in this area.

## 2. Materials and Methods

Sports nutrition and strength and conditioning practitioners working in professional sports were recruited to complete a survey aimed at investigating their use, practices and attitudes towards TC supplementation. Eligibility criteria included: (1) registration with a recognised professional body/organisation; (2) provision of dietary supplementation advice to athletes. The study was approved by the ethics committee of University of Exeter. Data were collected between May–August 2020.

The survey was distributed nationally and internationally via several sports nutrition and exercise science professional organisations and on multiple social media channels. Respondents completed and submitted the survey anonymously via an online survey software (Qualtrics, WA, USA). Access to the survey was provided after respondents answered two questions to confirm their eligibility and gave their written informed consent. The two questions assessed the eligibility criteria of registration with a recognised professional body/organisation and the provision of dietary supplementation advice to athletes.

The English language only questionnaire contained 28 questions divided into three sections and was designed to gather information on the demographics of the responders, and their use, practices and attitude towards TC supplementation. The questionnaire is included in the Appendix A. A combination of multiple choice (15), multiple option (10) and open ended (3) questions were included in the questionnaire. Data were included in the statistical analysis if a responder completed at least two out of the three sections of the questionnaire.

Section A of the questionnaire contained nine questions (5 multiple choice, 2 multiple option and 2 open ended questions) relating to the demographics of the respondents and one question identifying their recommendation of TC supplements. This section was completed by all respondents and gathered information on the age, gender, academic qualifications, support role, professional experience, sports in which they were professionally involved, job status and the age group of the athletes with which the respondents worked at the time of completing the questionnaire. Respondents were also asked to report their recommendations of TC supplements from the following answer options: “I currently recommend TC supplements to athletes”, “I previously recommended TC supplements, but no longer do so”, “I have not recommended so far TC supplements, but I plan on doing so in the future” and “I have never recommended TC supplements and I do not plan to do so”. This question was mandatory to all respondents.

Section B was completed by all respondents except for those who indicated that they have not used TC supplements and were not planning on doing so in the future. This section included ten questions (5 multiple choice and 5 multiple option) that investigated the use and practices of the respondents for TC supplementation. Specifically, these questions addressed the goals of the respondents when recommending TC supplements, the type of sports where the TC supplements were recommended, the type of TC supplements recommended, the duration of the supplementation protocol and the pre-loading protocol, the daily polyphenol dose used, the periods of the season when TC supplements were mostly used and the main information sources used by respondents when developing the supplementation protocol.

Section C was completed by all respondents and included nine questions (5 multiple choice, 3 multiple option and 1 open ended) that investigated the attitudes of the responders towards the evidence-base for benefits of TC supplementation and the main challenges faced by respondents when making decisions on their TC supplementation protocol, any negative effects noted when utilising TC supplements and any feedback received from athletes regarding the palatability of TC supplements. An open-ended question was used to identify any improvements noted by the practitioners or reported by athletes following TC supplementation. The answers collected for this question were subsequently categorised in different themes. This section also contained multiple choice questions that investigated the opinions of respondents towards the potential effects of TC supplements on training adaptations, the development of future research studies and TC supplements.

Data collected were analysed in SPSS Statistics (version 26.0, Chicago, IL, USA). Frequency analysis was conducted for all questions and the results are presented as percentage of responses. Chi-squared analysis was used to assess the effects of professional experience on TC recommendations.

## 3. Results

Thirty-five respondents met the eligibility criteria and completed at least two of three questionnaire sections. Respondents (33.4 ± 8.1 y, 66% male and 34% female) were sports nutritionist/dietitians (85%), strength and conditioning coaches (9%) and athletic performance coaches (6%). Respondents had varying degrees of professional experience (0–2 y, 34%; 2–5 y, 21%; 5–10 y, 23%; and 10+ y, 22%), and educational qualification (postgraduate degree, 74%; PhD, 18%; undergraduate degree, 8%). Respondents worked professionally in a range of sports: football/soccer (21%); rugby union (14%), athletics (13%), cycling (10%), tennis (9%), swimming (8%), other sports (25%), with 63% of the respondents working in at least two sports.

Forty-six percent of the respondents were currently recommending TC supplements, 11% had previously recommended TC supplements, 26% have not previously recommended TC supplements but plan on doing so in the future, and 17% of responders had not recommended TC supplements and were not planning on doing so in the future (Figure 1a). Improved exercise recovery and improved sleep duration and quality were the primary reasons practitioners recommended TC supplements (50% and 26% of total responses, respectively) with a smaller proportion of respondents seeking other outcomes (11% enhanced exercise performance, 9% improved immune function, and 4% enhanced training adaptations) (Figure 1b). TC supplements were most frequently recommended during periods of the season with multiple condensed athletic events/matches (50%) and during pre-season or demanding training blocks (24%) (Figure 1c). Eighteen percent of respondents indicated they recommend TC supplements “before and after every match/competition”, 4% indicated a daily use, whilst 4% of respondents were unsure about when to use TC supplements.

When the respondents recommended TC use, acute supplementation was the most common approach (28%), 15% of respondents indicated a ‘2–3 days’ supplementation period, 12% a duration of 5–8 days and 3% indicated a supplementation period of ‘3–5 days’ (Figure 2a). In contrast, 12% of respondents indicated a continuous use of the supplement. Twelve percent were “unsure” about the recommended duration, meanwhile 18% of responses indicated an “other” duration, described as a “daily use for the duration of a multi-days competition or demanding training camp/block with a 2–3 day pre-load” (*n* = 4). When using TC supplements for enhancing exercise recovery, 52% of respondents indicated that they were unsure about the duration of the pre-loading protocol (Figure 2b). Seventeen percent of respondents indicated that their recommended loading protocol started one day prior to competition, 14% and 11% indicated a 2-day and a 3-day pre-loading protocol, respectively, and only 3% indicated a loading protocol of 4–5 days and 6–8 days.

Fifty-two percent of respondents indicated that they were unsure of the recommended daily polyphenol dose with TC supplementation (Figure 2c). Twenty percent of respondents recommended 300–600 mg polyphenols·day^−1^, meanwhile 14% recommended 600–800 mg polyphenols/day. Only 7% of responders recommended 1000–1400 mg polyphenols·day^−1^, and the remaining 7% selected the answer option “other” without indicating any daily dosage.

Improved post-exercise recovery (34%) and improved sleep duration and quality (25%) were the most selected answers when the respondents indicated their viewpoint on the benefits of TC supplementation (Figure 3a). Improved endurance performance (9%), improved training adaptations (8%), improved immunity and general health (8%) and improved sprint performance (7%) were other indicated benefits of TC supplementation. The remaining 9% was represented by the answer option “others”, which included improved strength performance (*n* = 1), enhanced injury rehabilitation (*n* = 1), enhanced muscle mass development (*n* = 1) and unsure/do not have a strong opinion (*n* = 4). The main challenge highlighted by respondents when making decisions regarding TC supplementation was the mixed literature findings for the efficacy of this nutritional strategy (37%) (Figure 3b). The lack of an established effective supplementation protocol and the lack of research describing the effects of TC supplementation on training adaptations were both selected by 20% of respondents. Eight percent of respondents prioritised other supplements over TC, whilst the remaining 15% provided other answers: using a food-first approach (*n* = 2), budget concerns (*n* = 1) and difficulty in selecting a TC supplement from currently available TC supplements (*n* = 1).

Nineteen respondents answered the open question regarding improvements noticed with their athletes or feedback received from them following the use of TC supplements. The answer provided were categorised in the following three themes: improved sleep quality (*n* = 13) improved overall post-exercise recovery (*n* = 8), and reduced muscle soreness and improved overall wellbeing (*n* = 6).

## 4. Discussions

This was the first study to investigate the use of TC supplements in high performance athletic environments, as well as the practices and attitudes of sports nutrition and strength and conditioning practitioners towards this supplement. Over 55% of the respondents were currently recommending or have previously recommended TC supplements in their practice with athletes, meanwhile, an additional 26% were planning on recommending them in the future. More experienced practitioners tended to be more likely to recommend TC to their athletes, but views on the nature of potential TC ergogenic effects were not altered by experience. These findings indicate a potential high interest and use of TC supplements by sports nutrition and strength and conditioning practitioners across a large range of sports.

Improved exercise recovery (50%) and improved sleep duration and quality (26%) were the most indicated reasons the respondents recommended TC supplements. The supplements were most frequently recommended during periods with multiple condensed athletic events and during pre-season or demanding training blocks; periods when exercise recovery and sleep represent key priorities for athletes. Indeed, multiple studies have shown that exercise recovery is enhanced by TC supplementation across a wide range of exercise modalities [2,4,8], thus supporting this applied practice. However, to date, only one study has shown that TC supplementation may enhance sleep duration and quality in young healthy individuals [11]. Beneficial effects of TC supplementation on sleep have been reported in two other studies [12,25]; however, the participants were elderly and suffered insomnia, so applicability of these findings to competitive young athletes is unclear. There are currently limited data supporting the beneficial effects of TC supplements on enhancing sleep, this calls into question the relative high use of the supplement for this purpose by the respondents. Use of acute supplementation with TC for enhancing exercise performance was indicated in a relative low number of responses (11%). This is likely a reflection of the limited amount of research conducted on this topic so far, with only one study showing ergogenic effects for this strategy [13].

The duration of the supplementation protocols recommended by the respondents varied, with acute supplementation (regardless of goal) being recommended most frequently (28%). Acute supplementation may be sufficient for enhancing exercise performance if the TC dose is consumed 1.5–2 h prior to competition [13], however it may not be effective if consumed post-competition for enhancing exercise recovery, which was the most frequent goal indicated by respondents. Fifteen percent of responses indicated a supplementation duration of 2–3 days, which may also not be sufficient for enhancing exercise recovery. Although to date, no studies have directly compared the efficacy of different durations, improvements in exercise recovery were previously found when TC supplements were consumed for 7–10 days (4–7 days pre-load) [4,8,10]. In contrast, when the supplementation protocol lasted 3–5 days (1–2 days pre-load) [23,24], exercise recovery was not enhanced. Furthermore, when respondents were asked about the duration of the pre-loading protocol used when aiming to enhance exercise recovery, over 50% of respondents indicated “unsure”. Practitioners that recommended TC acutely for enhancing exercise recovery also selected “unsure”, as there was not a “no loading” answer option for this question. This limited knowledge of the appropriate duration of the pre-loading protocol is likely to be driven by the wide variety of supplementation protocols used within the literature and the lack of empirical evidence on which to base this decision. Sport-specific factors (culture, competition timetable, team line-up announcement, etc.) may also influence the duration of the supplementation protocols implemented by practitioners. Nevertheless, these findings indicate that practitioners require further education regarding the implementation of an effective TC supplementation protocol, but also the need for future studies examining this question.

Over 50% of respondents indicated that they were unsure about the quantity of daily total polyphenols to recommend as part of a TC supplementation protocol. This lack of clarity surrounding the quantity of polyphenols consumed daily through a TC supplementation protocol is also present within the literature where only a small number of studies conducted direct analysis of the TC supplement used [4,6,12]. Synthesis of previous research suggests a daily polyphenol dose of ~1200 mg may be required for exercise recovery to be enhanced [6], whilst 500–600 mg may be sufficient for enhancing exercise performance [14], but not for accelerating post-exercise recovery [26]. Further studies designed to establish optimal TC polyphenol supplementation doses for enhancing post-exercise recovery and exercise performance are thus required and are likely to play an important role in better informing practitioners.

This study also investigated the viewpoints of respondents on the literature-supported beneficial effects for TC supplementation. Improved exercise recovery and improved sleep duration and quality were the most frequent answers (34% and 25%, respectively). All respondents indicated at least one-literature supported benefit for TC supplementation, with the majority selecting at least two. These findings likely indicate that the high use and planned use of TC supplements amongst the respondents is driven by their understanding of the current body of literature. The main challenges indicated by the respondents when making decisions regarding TC supplementation included the mixed findings in the literature regarding the efficacy of the supplement (37%), the lack of an established supplementation protocol (20%) and the lack of studies illustrating its effects on training adaptations (20%). Further research is required to assuage these concerns and ensure selection of effective evidence-based supplementation strategies.

The respondents indicated that improved sleep quality, improved post-exercise recovery and reduced muscle soreness and improved overall wellbeing were the beneficial effects reported by athletes following TC supplementation. These observations, while anecdotal, are partly in line with previous literature showing improved exercise recovery and reduced muscle soreness with TC supplementation [7,8]. These reported beneficial effects closely matched the most indicated goal of the practitioners when using TC supplements: improved exercise recovery. Nevertheless, the most indicated beneficial effect reported by respondents was improved sleep quality (*n* = 13). To date, however, only one study investigated this topic and found beneficial effects for TC supplementation in young individuals [11]. This discrepancy between practice-based anecdotes and the current body of literature emphasises the need for future well-controlled studies to determine the effects of TC supplementation on sleep quality and duration in this population.

Lastly, although the findings indicate a high use and interest in TC supplementation from these sports nutrition and strength and conditioning practitioners, it is possible that practitioners who were aware of the potential benefits associated with this supplement were more likely to engage in the study. Furthermore, although the number of respondents was similar to previous studies investigating applied practices of sport science related practitioners [27,28], the small sample size provided limited opportunity for further analysis beyond descriptive statistics. Future larger scale studies could investigate whether the use and practices for TC supplementation differ between sports nutrition and strength and conditioning practitioners, or whether these are influenced by the sport(s) in which the practitioners are working.

## 5. Conclusions

A high proportion of sports nutrition and strength and conditioning practitioner respondents were recommending or were planning on recommending TC supplements. Improved exercise recovery and improved sleep duration and quality were the main goals of the practitioners when recommending TC supplements. The duration and the doses recommended as part of the supplementation protocols varied substantially between respondents and seldom matched the protocols shown to be effective within the literature. The main challenges indicated by respondents were the conflicting research evidence regarding the effectiveness of TC supplements, the lack of consensus on an optimal supplementation protocol and the lack of research illustrating the effects of TC supplementation on training adaptations. Future studies that address these concerns should be conducted.

## Figures and Tables

**Figure 1 sports-09-00002-f001:**
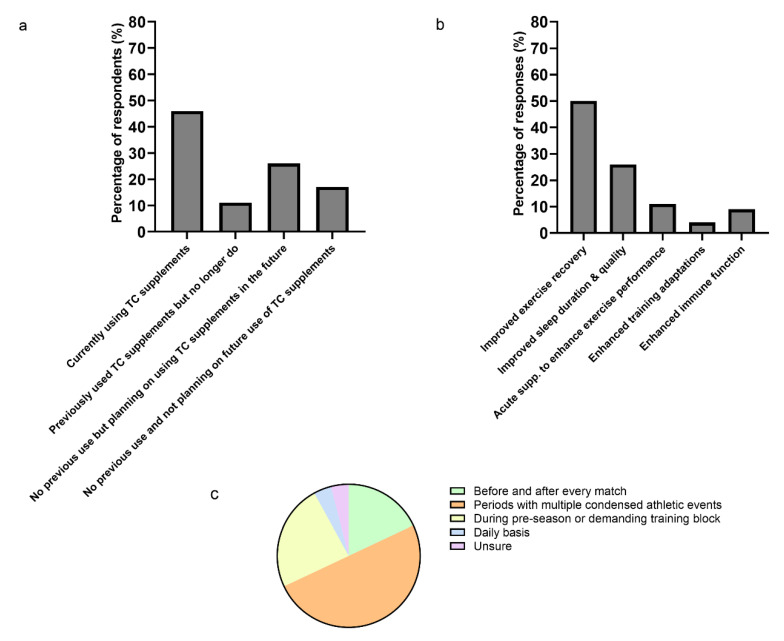
(**a**) Presents the use of tart cherry (TC) supplements among the respondents, *n* = 35. (**b**) Presents the main goals of the responders when using TC supplements. Respondents were able to provide multiple answers to this question (total answers = 54). (**c**) Presents the periods of the season when the responders used TC supplements. Respondents were able to provide multiple answers to this question (total answers = 46).

**Figure 2 sports-09-00002-f002:**
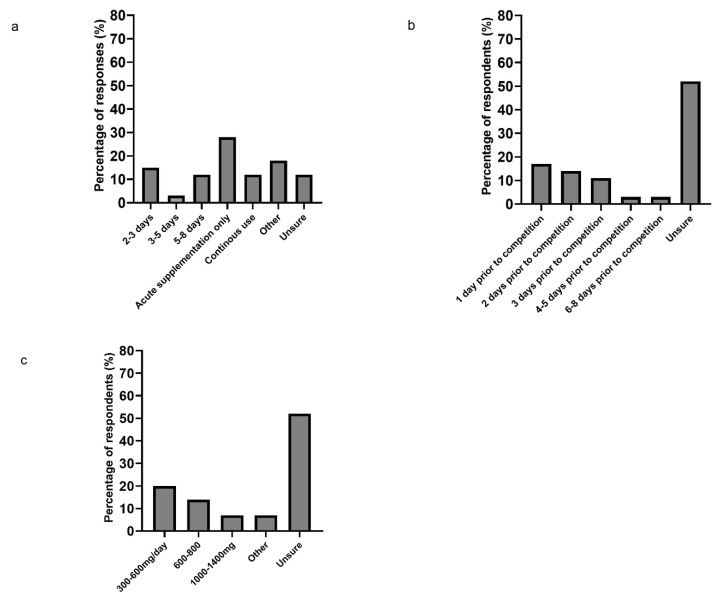
(**a**) Presents the duration of the usual TC supplementation protocol used by respondents. Respondents were able to provide multiple answers to this question (total answers = 34). (**b**) Presents the pre-load duration of the supplementation protocol when TC is used for enhancing exercise recovery, *n* = 29. (**c**) Presents the daily polyphenol dose recommended by respondents, *n* = 29.

**Figure 3 sports-09-00002-f003:**
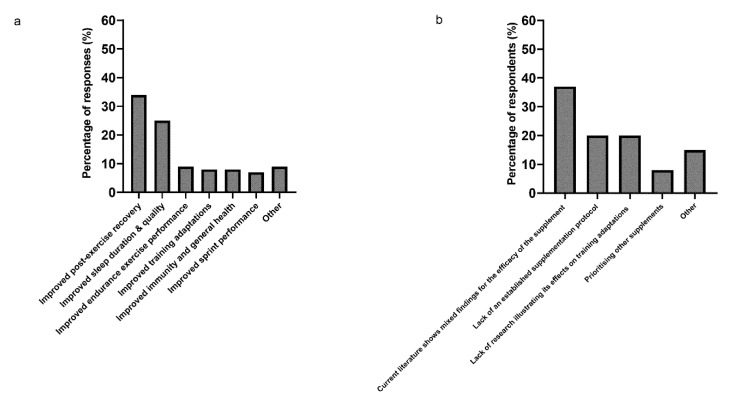
(**a**) Presents viewpoint of the respondents on the literature-supported benefits of TC supplementation. Respondents were able to provide multiple answers to this question (total answers = 76). (**b**) Presents the main challenges faced by respondents when deciding on whether and how to implement a TC supplementation protocol, *n* = 35.

## Data Availability

No new data were created or analyzed in this study. Data sharing is not applicable to this article.

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
