# Peer review of "Use, Practices and Attitudes of Sports Nutrition and Strength and Conditioning Practitioners towards Tart Cherry Supplementation"

_sports, 2020, doi:10.3390/sports9010002_

Round 1

Reviewer 1 Report

The study exams practicing sports nutrition and strength & conditioning coaches frequency of suggesting the use of tart cherry. Although the study had a relatively low number of respondents it was able to give insight into current recommendations made by practicing professionals in the field. A high number of respondents had post-graduate degrees. Although this has become a norm in both sports nutrition and strength & conditioning as they become more professionalized. It also questions if educated professionals are more willing to participate in research studies. 

Graphs and figures would be easier to interpret with titles above each graph. The reader is forced to look down at the caption to understand the graphs. This distracts the reader from the author's point. 

Line 133 - More experienced practitioners tended “More” is not needed at the start of the sentence.

The authors do not go into the mechanisms of action of how tart cherry juice and polyphenols wound benefit athletes but that is not the point of this line of research. The research focuses on sports nutrition and strength & conditioning recommendation on using tart cherry juice. Lines 187-190 that discuss the open-ended questions on the survey are nothing more than anecdotal accounts from practitioners. These accounts are not from the client/athletes that would have been taking the tart cherry supplements. These second-hand accounts add very little to the main focus of this line of research and should be excluded.  

Overall the study is well designed and gives valuable insight into the tart cherry supplementation recommendations made by sports nutrition and strength & conditioning professionals. The research also suggests gaps in the current practice of professionals and the scientific data with regards to the timing of supplementation and dosage. 

Reviewer 2 Report

Abstract and Introduction sections are nicely done without comments.

How did the respondents confirm their eligibility?

How did the authors make the statistical analysis of the respondents’ data if they included some respondents with 2/3 of answered surveys? Please explain.

35 qualified respondents is rather small sample to make some conclusions. It should be much bigger.

Line 133/134 - "More experienced practitioners tended to be more likely to recommend TC use (p=0.064)." This is not statistically significant. How did the authors conclude that?

In the aim of the study the authors mentioned that they will investigate attitudes of sports nutrition. In Results and Discussion sections the authors did not mention results regarding attitudes, so they should correct either aim or results/discussion.
